# Recommended Physiotherapy Modalities for Oncology Patients with Palliative Needs and Its Influence on Patient-Reported Outcome Measures: A Systematic Review

**DOI:** 10.3390/cancers16193371

**Published:** 2024-10-01

**Authors:** Luna Gauchez, Shannon Lauryn L. Boyle, Shinfu Selena Eekman, Sarah Harnie, Lore Decoster, Filip Van Ginderdeuren, Len De Nys, Nele Adriaenssens

**Affiliations:** 1Physiotherapy Human Physiology and Anatomy Department (KIMA), Vrije Universiteit Brussel, Laarbeeklaan 103, 1090 Brussels, Belgium; luna.gauchez@vub.be (L.G.); shannon.lauryn.l.boyle@vub.be (S.L.L.B.); shinfu.selena.eekman@vub.be (S.S.E.); sarah.harnie@vub.be (S.H.); filip.van.ginderdeuren@vub.be (F.V.G.); len.de.nys@vub.be (L.D.N.); 2Rehabilitation Research, Vrije Universiteit Brussel, Laarbeeklaan 103, 1090 Brussels, Belgium; 3End-of-Life Care Research Group, Vrije Universiteit Brussel & Ghent University, Laarbeeklaan 103, 1090 Brussel, Belgium; lore.decoster@vub.be; 4Medical Oncology Department, Universitair Ziekenhuis Brussel, Laarbeeklaan 101, 1090 Brussels, Belgium; 5Physical Medicine and Rehabilitation, Universitair Ziekenhuis Brussel, Laarbeeklaan 101, 1090 Brussels, Belgium

**Keywords:** physiotherapy, physical therapy, physiotherapist, physical therapist, neoplasms, palliative care, patient-reported outcome measures, quality of life

## Abstract

**Simple Summary:**

This review explores the role of physiotherapy in early and traditional palliative care (PC) for cancer patients, focusing on its effects on patient-reported outcomes (PROMs) such as fatigue, pain, and quality of life (QoL). The findings suggest that certain physiotherapy interventions, like cognitive behavioral therapy (CBT) and massage, can improve these outcomes, while others show limited or no significant benefits. However, the overall quality of the studies reviewed was low, highlighting the need for more rigorous research in this area to better understand how physiotherapy can support patients in PC.

**Abstract:**

Background: This review aims to explore the role of physiotherapy in early and traditional palliative care (PC) for oncology patients, focusing on its impact on six patient-reported outcomes (PROMs), namely fatigue, pain, cachexia, quality of life (QoL), physical functioning (PHF), and psychosocial functioning (PSF). The purpose is to assess the effectiveness of various physiotherapy interventions and identify gaps in the current research to understand their potential benefits in PC better. Methods: A systematic literature search was conducted across PubMed, Embase, and Web of Science, concluding on 21 December 2023. Two independent reviewers screened the articles for inclusion. The Cochrane Risk of Bias Tool 2 was employed to assess the risk of bias, while the GRADE approach was used to evaluate the certainty of the evidence. Results: Nine randomized controlled trials (RCTs) were included, with most showing a high risk of bias, particularly in outcome measurement and missing data. Cognitive behavioral therapy (CBT) was the only intervention that significantly reduced fatigue, enhanced PHF, and improved QoL and emotional functioning. Graded exercise therapy (GET) did not yield significant results. Combined interventions, such as education with problem-solving or nutritional counseling with physical activity, showed no significant effects. Massage significantly improved QoL and reduced pain, while physical application therapies were effective in pain reduction. Mindful breathing exercises (MBE) improved QoL but had a non-significant impact on appetite. The overall certainty of the evidence was low. Conclusions: Physiotherapy can positively influence PROMs in oncology PC; however, the low quality and high risk of bias in existing studies highlight the need for more rigorous research to confirm these findings and guide clinical practice.

## 1. Introduction

Cancer, defined by the World Health Organization (WHO) as a disease characterized by the rapid growth of abnormal cells that can invade and spread to other parts of the body, is one of the top six most common causes of death worldwide [1,2]. In 2022, the WHO reported approximately 20 billion new cancer cases and 10 billion cancer-related deaths, with a projected 76.6% increase in new cases by 2050 [1,2,3,4]. In Belgium, cancer affects a significant portion of the population, with one in three men and one in four women diagnosed before the age of 75, and most cases occurring in individuals over 60 years old [5]. Cancer is the leading cause of death in Belgium, responsible for 24.2% of all deaths in 2021 [6].

Palliative care (PC) for cancer patients can be categorized as either generalist or specialist. Generalist PC is generally administered by healthcare professionals already engaged in the patient’s care, such as general practitioners and nurses. In contrast, specialist PC requires a multidisciplinary approach that ideally includes a range of healthcare providers specialized in PC to address complex needs [7]. However, in Belgium, the integration of such multidisciplinary teams, particularly the inclusion of physiotherapists, within specialist PC is not yet fully established [8,9,10,11,12,13,14,15,16].

Physiotherapy (PT) in PC aims to enhance physical and functional well-being and manage symptoms through methods like therapeutic exercise, manual therapy, and patient education [10,11,12,13,14,15,16]. PT in PC oncology is increasingly recognized for its potential to improve patients’ QoL, particularly in managing symptoms such as pain, fatigue, and mobility limitations [14]. It aims to enhance physical function and independence, even in advanced stages of cancer. However, despite its benefits, the integration of physiotherapy into oncology care plans remains inconsistent across healthcare settings. Barriers such as limited awareness among healthcare providers, inadequate funding, and variability in training programs contribute to the underutilization of physiotherapy in palliative care. While some guidelines advocate for its inclusion, more research is needed to establish standardized protocols and define the physiotherapist’s role within interdisciplinary palliative care teams.

Early integrated specialized palliative care (EISPC) represents the current standard, integrating PC in the disease trajectory and alongside curative treatments. This approach aims to enhance patient outcomes by addressing symptoms and providing comprehensive support from the outset [17]. In contrast, traditional PC is typically introduced later, focusing mainly on end-of-life care. Although EISPC has shown promising results in improving patient outcomes, there is still a notable gap in evidence concerning the role of PT in both EISPC and traditional PC settings [16,17].

Given the global prevalence of cancer and the current lack of evidence about the role of PT in PC, this systematic review seeks to clarify the contributions of physiotherapists in both early and traditional PC for oncology patients. It will specifically examine how PT affects key patient-reported outcomes (PROMs). Gaining a clearer understanding of this role is essential for improving patient care and outcomes in this vulnerable population.

## 2. Materials and Methods

This systematic review was conducted to evaluate the effectiveness of PT interventions in palliative oncology care compared to usual care (UC), following the PRISMA guidelines [18].

It was conducted by following the Preferred Reporting Items for Systematic Reviews and Meta-Analyses (PRISMA). This study was also registered in the International Prospective Register of Systematic Reviews.

### 2.1. Study Selection Procedure

The databases used for the literature search were PubMed, Web of Science, and EMBASE. A PICO framework was employed to construct the search strategy, focusing on:Population: adults aged 18 and over with solid tumors, with more than 50% of the sample having palliative needs.Intervention: PT, including various modalities beyond just the term “physiotherapy” to capture a comprehensive range of interventions.Comparison: usual care (UC).Outcome: six patient-reported outcomes (PROMs): fatigue, quality of life (QoL), nutrition, pain, psychosocial functioning (PSF), and PHF.

The search terms and strategy details are available in Appendix A. The search period concluded on 24 September 2023, with all articles in the database being considered, regardless of their publication date. The final search was conducted on 21 December 2023. Additionally, a relevant paper provided by a professor was included.

The selection process involved three stages and was conducted by two blinded assessors, with a third assessor consulted in cases of uncertainty. Initially, duplicates were removed using Endnote X9 and the Rayyan web application (https://www.rayyan.ai, accessed on 1 January 2024). The remaining 1990 articles were screened based on titles and abstracts. Articles were excluded if they: (1) Were not in English, Dutch, or French. (2) Did not involve Randomized Controlled Trials (RCTs). (3) Involved populations under 18, non-oncology patients, patients with liquid tumors, or less than 100% of patients without palliative needs. (4) Did not describe PT modalities or UC in sufficient detail. (5) Focused on outcomes other than the specified PROMs.

In total, 1948 articles were excluded. In the final stage, 43 full-text articles were assessed for eligibility, and nine articles were included. Detailed eligibility criteria are outlined in Table 1, and the selection process is illustrated in the PRISMA flowchart (Figure 1).

### 2.2. Data Extraction

Data were extracted based on the PICO dimensions using a custom Excel spreadsheet, which was piloted with one article before use. The data extraction was performed independently by two assessors, who then compared and discussed their findings to ensure accuracy.

### 2.3. Assessment of Risk of Bias (RoB)

The RoB was evaluated using the Cochrane RoB Tool 2 (RoB 2), which includes five domains: (1) bias from the randomization process, (2) bias due to deviations from intended interventions, (3) bias due to missing outcome data, (4) bias in outcome measurement, and (5) bias in selection of the reported results. This assessment was conducted independently by two blinded assessors, with subsequent comparison and discussion to resolve discrepancies.

## 3. Results

### 3.1. Study Characteristics

Table 2 provides detailed information on the sample sizes, drop-out rates, and retention rates for each study, while Table 3 summarizes additional study characteristics, including tumor type, patient demographics, study settings, interventions, and outcomes assessed. Both tables are organized from the oldest to the most recent publication date, facilitating a clearer understanding of the progression and context of the research over time.

The included studies predominantly involved palliative oncology patients diagnosed with solid tumors. Sample sizes ranged from 37 to 380 participants, with mean ages spanning from 40 to 70 years. The majority of studies were conducted within traditional PC settings, with one study situated in an EISPC context. Interventions across the studies varied, encompassing breathing exercises, educational programs, and physical therapy, while control groups typically received standard care, such as chemotherapy, radiation therapy, or pain management. The primary outcomes assessed included fatigue, quality of life (QoL), nutritional status, physical health function (PHF), psychosocial function (PSF), and pain. Statistical significance was defined as a *p*-value of less than 0.05.

Table 4 presents a comprehensive summary of the primary findings, with studies systematically grouped according to the outcomes they assessed. Studies that investigated identical outcomes—fatigue, quality of life (QoL), nutritional status, physical health function (PHF), psychosocial function (PSF), and pain—are consolidated to allow for direct comparison.

### 3.2. Fatigue

Look et al. (2021) investigated the effect of mindful breathing exercises (MBE) and found a significant reduction in drowsiness in the intervention group, though no significant difference in tiredness was observed [26]. Maddocks et al. (2013) assessed neuromuscular electrostimulation (NMES) and found no significant changes in fatigue dimensions, although mental fatigue was significantly higher in the control group [22]. Poort et al. (2020) compared cognitive behavioral therapy (CBT) and graded exercise therapy (GET) with usual care, finding significant reductions in fatigue for the CBT group but not for GET [25]. Uster et al. (2018) observed a reduction in fatigue in the intervention group, though results were not statistically significant [23].

### 3.3. QoL

Bakitas et al. (2009) found that educational and problem-solving sessions improved QoL, though not significantly [21]. Notably, this study is the only RCT included in this section that demonstrates a low risk of bias, highlighting its reliability in evaluating QoL outcomes. Kutner et al. (2008) reported significant improvements in QoL with massage compared to control [20]. Maddocks et al. (2013) found no significant differences in QoL between NMES and control groups, with some improvement in specific domains [22]. Nottelmann et al. (2021) observed significant improvements in QoL at 12 weeks with educational and PT sessions [27]. Poort et al. (2020) found CBT effective in improving QoL at 14 weeks, while GET did not show significant differences [25]. Uster et al. (2018) showed non-significant improvements in QoL in both groups [23].

### 3.4. Nutrition

Look et al. (2021) found a significant improvement in appetite with MBE [26]. Uster et al. (2018) reported no significant changes in daily protein intake or body weight, though the intervention group experienced less decline compared to the control group [23].

### 3.5. PHF

Maddocks et al. (2013) found no significant changes in quadriceps muscle strength, with a greater reduction in the control group [22]. Poort et al. (2020) reported significant improvements in PHF with CBT [25]. Uster et al. (2018) observed improvements in handgrip strength and the six minutes walk test (6MWT) in the intervention group, though not statistically significant [23].

### 3.6. PSF

Bakitas et al. (2009) found enhanced mood in the intervention group [21]. Massage showed significant improvements in mood compared to control [20]. Look et al. (2021) reported significant reductions in anxiety and depression with MBE [26]. Poort et al. (2020) found CBT improved emotional functioning significantly compared to usual care [25]. Uster et al. (2018) showed non-significant improvements in emotional functioning in the intervention group [23].

### 3.7. Pain

Kashyap et al. (2020) found significant pain reduction with scrambler therapy compared to usual care [24]. Massage also resulted in significant pain reduction [20]. Look et al. (2021) showed a decrease in pain in the intervention group, though not statistically significant [26]. Tsai et al. (2007) reported significantly lower pain intensity with electromyography biofeedback [21]. Uster et al. (2018) found increased pain in both groups, with a greater increase in the control group, but results were not statistically significant [23].

### 3.8. GRADE

The certainty of evidence for all outcomes was very low due to factors such as lack of blinding, reliance on patient-reported outcomes, and insufficient reporting of confidence intervals and effect sizes. The RoB analysis and summary can be found in Figure 2 and Figure 3.

### 3.9. Summary

Across multiple outcomes, PT consistently demonstrated positive effects, particularly in reducing fatigue, improving QoL, and enhancing PSF. Notably, four RCTs reported significant improvements in fatigue, while six RCTs showed improvements in QoL. However, the overall certainty of evidence was rated as very low, largely due to heterogeneity in interventions and study designs.

In PHF, there was a notable divergence between subjective reports, which generally indicated improvements, and objective measurements, which produced mixed results. This contradiction highlights the inherent complexity in accurately measuring physical function within this population.

Additionally, the evidence suggests modest improvements in pain and nutrition, although the certainty of these findings remains low. Pain improvements were observed in five RCTs, but further high-quality research is needed to substantiate these effects and strengthen the evidence base across all outcomes.

## 4. Discussion

### 4.1. Interpretation of Findings

This systematic review aimed to identify effective PT modalities for palliative oncology patients and their impact on fatigue, QoL, nutrition, PHF, PSF, and pain. The findings reveal that certain interventions, including MBE and CBT, demonstrate effectiveness in reducing fatigue and improving QoL. Notably, these results align with previous studies indicating that psychological and mind-body interventions can significantly address cancer-related fatigue (CRF) and enhance patients’ overall well-being [25,26].

The effectiveness of MBE and CBT in reducing fatigue is supported by their ability to address both psychological and physiological aspects of CRF. This is consistent with the literature suggesting that fatigue in oncology patients is often multifactorial, involving psychological distress, anemia, and pain [28,29,30]. MBE’s success in alleviating fatigue may be attributed to its focus on relaxation and stress reduction, while CBT’s impact is likely due to its cognitive restructuring techniques and behavioral strategies [25,26]. Furthermore, interventions such as NMES and GET combined with PT also show promise in mitigating fatigue, though the evidence is less robust compared to MBE and CBT [25,26].

QoL, a critical outcome in PC, was found to be positively influenced by several interventions, including massage, CBT, and educational sessions combined with exercise [20,25,27]. This is particularly important given the clinical and economic implications of improving QoL in PC settings. The improvements in QoL observed in this review are in line with previous research highlighting the benefits of holistic approaches that integrate physical and psychological interventions [31,32,33]. Educational and problem-solving sessions, NMES, and nutritional counseling were also associated with QoL improvements, further emphasizing the multifaceted nature of effective PC [21,22,23].

Nutrition, a key component in managing cachexia and malnutrition in oncology, and especially palliative patients, was positively impacted by MBE and nutritional counseling combined with exercise. Improved appetite and daily protein intake observed with these interventions underscore the importance of addressing nutritional needs in conjunction with other therapeutic strategies [23,26]. These findings are consistent with existing literature on the role of nutritional and physical interventions in managing cachexia and its related symptoms [34,35,36].

PHF and PSF were also positively influenced by PT. Exercise interventions, including resistance training and aerobic exercise, were found to maintain or improve physical functionality, which is crucial given the impact of physical decline on patient outcomes [37,38,39,40]. The observed benefits of PT on PHF align with studies suggesting that physical activity can mitigate functional decline and improve QoL [8,37,40,41]. CBT’s role in improving PSF further supports the notion that addressing emotional and cognitive aspects can enhance overall well-being [25].

Pain management in palliative oncology is essential, given its high prevalence and impact on patient QoL. This review identified scrambler therapy, massage, and electromyography biofeedback-guided relaxation as effective in reducing pain intensity [20,21,22]. These findings are consistent with previous research demonstrating the efficacy of these interventions in managing pain, highlighting the need for a multimodal approach to pain relief [42,43,44,45,46].

### 4.2. Broader Context and Future Research

The findings of this review should be considered in the broader context of palliative oncology care. PT interventions, when combined with other modalities such as psychological and nutritional support, offer a comprehensive approach to managing the multifaceted needs of palliative patients. This aligns with the growing emphasis on integrated care models in palliative settings [47,48,49,50].

Future research should address several limitations identified in this review. The high risk of bias (RoB) in most studies, primarily due to reliance on self-reported outcomes and high dropout rates, underscores the need for more rigorous research designs. Attrition bias and small sample sizes in the included studies also reduce the statistical power of the results, complicating the ability to draw robust conclusions. Furthermore, the heterogeneity of interventions—PT encompasses various techniques such as CBT, GET, and massage—makes it challenging to generalize findings across diverse approaches. This variability may exaggerate perceived effectiveness, particularly given the overall low quality of the evidence.

To enhance the quality of future research, validated assessment tools should be consistently employed, and the impact of PT interventions should be considered within the context of multidisciplinary care [47,48,49]. Additionally, more research is needed to clarify the efficacy of specific interventions and their optimal combinations to enhance patient outcomes.

Moreover, the small number of studies included in this review highlights the need for further investigation into PT modalities in PC. Larger and more diverse studies are needed to confirm the effectiveness of various interventions and to explore their impact on different patient populations. Research should also investigate the long-term benefits and sustainability of these interventions, as well as their cost-effectiveness in the context of increasing demand for PC services [32,33].

### 4.3. Strengths and Limitations of the Review

This review adheres to the PRISMA guidelines, ensuring a systematic and reproducible methodology. The inclusion of studies from multiple databases and languages increases the comprehensiveness of the review. However, the small number of included studies and the limitations related to methodological quality and intervention scope pose challenges. Despite these limitations, the review provides valuable insights into the potential benefits of PT in managing key outcomes for palliative oncology patients and highlights areas for future research.

In summary, this systematic review contributes to the understanding of PT interventions in palliative oncology care. The findings underscore the importance of a multifaceted approach that integrates physical, psychological, and nutritional support to address the complex needs of this population. Future research should build on these findings to refine and enhance PC practices.

## 5. Conclusions

Based on the results of this systematic review, it can be concluded that PT is beneficial for improving both physical and psychological outcomes in oncology patients with palliative needs. Specifically, MBE, CBT, and massage have shown positive and significant effects on most outcomes, including fatigue, PSF, PHF, QoL, and pain. Additionally, physical interventions and educational sessions combined with PT show promising results for outcomes such as pain and QoL.

However, many studies did not demonstrate significant results, suggesting that while these interventions can be considered, they cannot be strongly recommended at this time. There is a significant gap in the literature regarding PT for this palliative oncology population, and the quality of the studies in this field is often insufficient. Therefore, PT interventions cannot be universally recommended. Future research should focus on conducting high-quality studies to explore which aspects of PT can be recommended for palliative oncology patients.

## Figures and Tables

**Figure 1 cancers-16-03371-f001:**
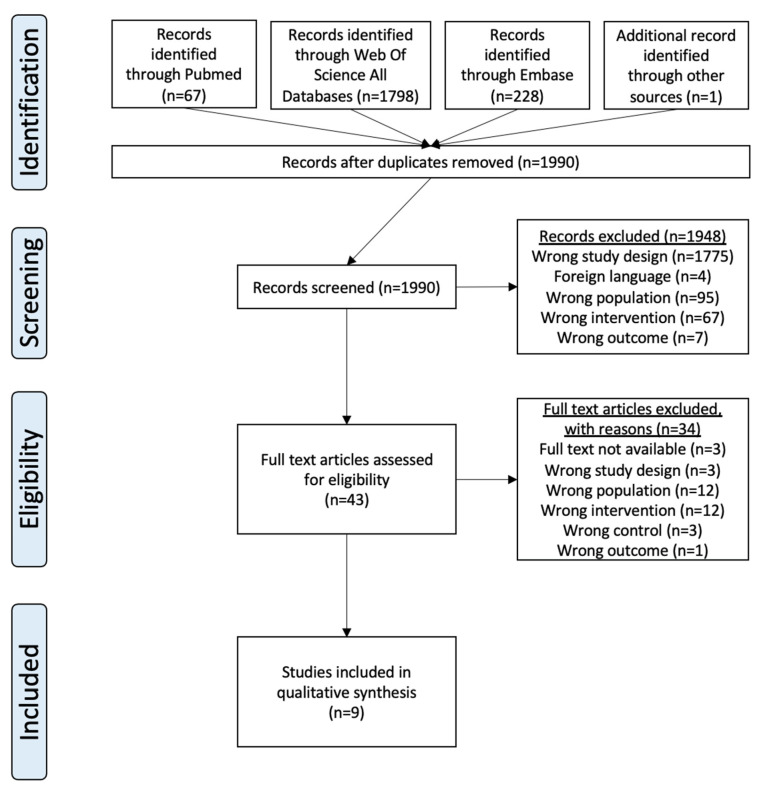
PRISMA Flowchart. This PRISMA 2020 flowchart is based on the PRISMA 2020 statement and has been adapted in accordance with the Creative Commons Attribution 4.0 (CC BY 4.0) license. Source: Page et al. (2021), PRISMA 2020 statement [18].

**Figure 2 cancers-16-03371-f002:**
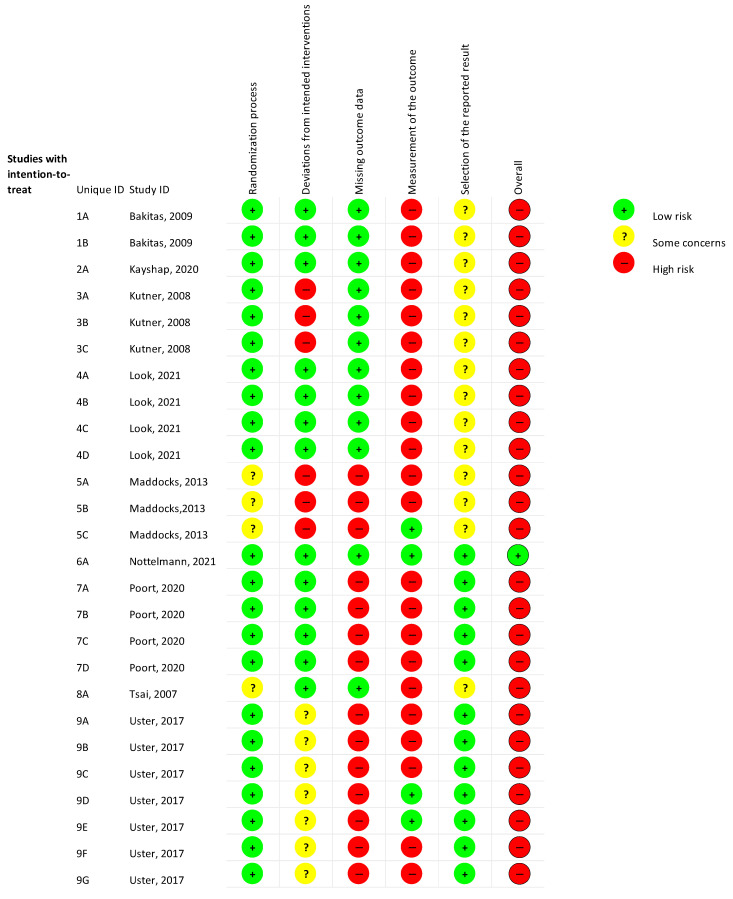
RoB [19,20,21,22,23,24,25,26,27].

**Figure 3 cancers-16-03371-f003:**
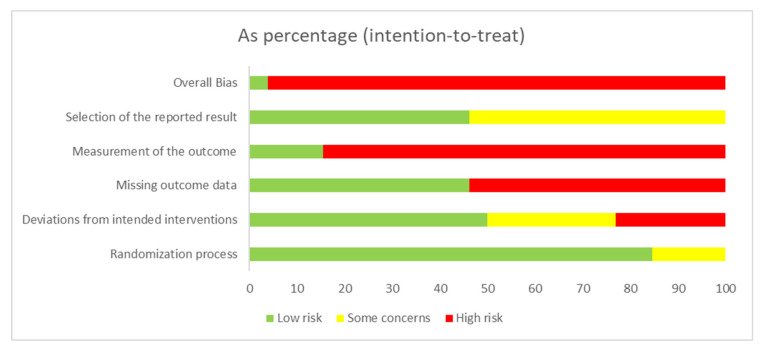
Summary RoB—as a percentage.

**Table 1 cancers-16-03371-t001:** In- & exclusion criteria.

Inclusion	Exclusion
Population - Solid tumour oncology patients with palliative needs - Age: adult (>18 y.o.) - 100% oncology patients with palliative needs	Population - Age: children and adolescents (<18 y.o.) - Non-oncology patients - Oncology patients without solid tumours - Oncology patients without palliative needs - <100% oncology patients without palliative needs
Intervention - Specification of type, intensity, frequency, duration of physical therapy program (min. 3 out of 4 criteria) - Scope of a physiotherapist: according to the Koninklijk Nederlands Genootschap voor Fysiotherapie (KNGF) and NICE guidelines	Intervention - Non-PT modalities - Aspects of PT not specified (more than 1 criterion not specified) - Interventions not within the defined scope of a physiotherapist
Comparison - Other PT programs or other PT programs in combination with medical treatment or medical treatment only	Comparison - Other programs than PT (in combination or without medical treatment) or medical treatment only
Outcome - Patient-reported outcome measures: fatigue, QoL, nutrition, pain, psychosocial or PHF	Outcome - Other patient-reported outcome measures than fatigue, QoL, nutrition, pain, psychosocial or PHF
Study design - Randomized controlled trials	Study design - Other study designs than randomized controlled trials
Other - Articles written English, Dutch, and French - Full text of articles available	Other - Other languages than English, Dutch, or French - No full text available

**Table 2 cancers-16-03371-t002:** Sample size, dropout and retention rate.

Author, Year	Sample Size Total (Baseline)	Sample Size I (Baseline)	Sample Size C (Baseline)	Retention I (%)	Retention C (%)	Dropout Total (%)	Mean Age ± SD I	Mean Age ± SD C
Tsai et al. (2007) [19]	*n* = 37	*n* = 20	*n* = 17	60%	70.59%	35.14%	≤40 (16.7)— 41–50 (58.3)— 51–60 (8.3)— ≥61 (16.7)	≤40 (8.3)— 41–50 (25)— 51–60 (25)— ≥61 (41.7)
Kutner et al. (2008) [20]	*n* = 380	*n* = 188	*n* = 192	80.32%	76.56%	21.58%	65.2 ± 14.4	64.2 ± 14.4
Bakitas et al. (2009) [21]	*n* = 322	*n* = 161	*n* = 161	90.06%	82.23%	13.35%	65.4 ± 10.3	65.2 ± 11.7
Maddocks et al. (2013) [22]	*n* = 49	*n* = 30	*n* = 19	50%	68.42%	42.86%	70	68
Uster et al. (2017) [23]	*n* = 58	*n* = 29	*n* = 29	82.76%	68.97%	24.14%	64.0 ± 11.0	62.0 ± 9.3
Kashyap et al. (2020) [24]	*n* = 80	*n* = 40	*n* = 40	97.50%	100%	1.25%	52 ± 9.98	47.4 ± 11.22
Poort et al. (2020) [25]	*n* = 134	CBT*n* = 46 & GET*n* = 42	*n* = 46	GET: 78.57% CBT: 84.78%	86.96%	16.42%	GET: 60.67 ± 10.75 CBT: 63.50 ± 8.15	63.93 ± 8.98
Look et al. (2021) [26]	*n* = 40	*n* = 20	*n* = 20	100%	100%	0%	66.75 ± 3.22	69.20 ± 2.54
Nottelmann et al. (2021) [27]	*n* = 301	*n* = 149	*n* = 139	24.83%	22.30%	26.91%	66 ± 9	66 ± 10

**Table 3 cancers-16-03371-t003:** Study characteristics.

Study (Author & Year)	Patient Population	Intervention	Components of the Intervention	Control	Outcomes	Traditional PC/EISPC	Type of PT Modality
Tsai et al. (2007) [19]	Adult patients with advanced cancer who scored 3 on the Brief Pain Inventory, KPS: 40–90	EMG biofeedback-assisted relaxation (first 2 sessions visual display, last 4 sessions closed eyes)	45′/sessions 3 × 3–7 min trials, 20% reduction in the EMG from pretraining level for 50% of the time in each trial, 6 sessions, 4 weeks	Usual care	PSF	Traditional PC	Relaxation Physical applications
Kutner et al. (2008) [20]	Advanced cancer adult patients with moderate pain, stages III-IV	Massage: gentle effleurage, petrissage, myofascial trigger point release	Session = 30′ effleurage (65% time), petrissage (35% time) & myofacial trigger point release (n = 3/session), up to 6 sessions with 24 h interval, 2 weeks	Control exposure: bilateral placement of hands, 3 min/location	QoL PSF Pain	Traditional PC	Massage
Bakitas et al. (2009) [21]	Patients with life-limiting cancer and new diagnosis within 8–12 w of GI tract, lung, genitourinary tract, breast cancer stage III or IV	Structured educational & problem-solving sessions (case-management, educational approach to encourage P activation, self-management and empowerment) + SMAs	Session 1: ±41′ Sessions 2–4: 30′ 4 sessions 1×/week	Usual care	QoL PSF	Traditional PC	Education
Maddocks et al. (2013) [22]	Adults with advanced (stage IV) NSCLC from thoracic oncology clinics scheduled to receive first line palliative chemotherapy	Neuromuscular electrical stimulation	Session = 30′ symmetrical biphasic squared pulses at 50 Hz, 350 microsec pulse width, duty cycle increasing on weekly basis from 11% to 18% to 25% and constant thereafter, daily (min 3×/week), 3 cycles chemo: 8 w 4 cycles chemo: 11 w	Usual care	Fatigue QoL PHF	Traditional PC	Physical applications
Uster et al. (2017) [23]	Patients with metastatic or locally advanced tumors of the gastrointestinal or the lung tracts + ECOG ≤ 2	Nutrition and physical intervention (group 2–6 pers): warm-up, strength & balance training	Session = 60′ Strength: 60–80% 1RM in 2 sets of 10 reps → increase R Balance: 1′ → 2′, 2×/week, 3 months	Usual care	Fatigue QoL Nutrition PHF PSF	Traditional PC	Exercise
Kashyap et al. (2020) [24]	Head, neck & thoracic cancer patients with NRS-11 more than 4, stage II-IV	Scrambler therapy (electrodes according dermatome)	40′ Scrambler therapy, intensity increased gradually, 5×/week, 2 weeks	Pain medication (WHO) + usual care	Pain	Traditional PC	Physical applications
Poort et al. (2020) [25]	Adult palliative cancer patients with severe cancer-related fatigue	CBT or GET (graded aerobic and resistance training)	CBT: 1 h sessions and GET: 2 h sessions, CBT: 10 individual sessions GET: 2×/week, 12 weeks	Usual care: guidelines by Netherlands Comprehensive Cancer organisation	Fatigue QoL PHF PSF	Traditional PC	CBT & GET (Education) (Exercise)
Look et al. (2021) [26]	Adult cancer patients with at least 5/10 on ESAS, ECOG I-IV	Mindful breathing exercise	20′ in 4 steps (5′ per step), once, one session	Usual care	Fatigue Nutrition PSF Pain	Traditional PC	Breathing exercises
Nottelmann et al. (2021) [27]	Adult cancer patients receiving systemic medical treatment for metastatic or unresectable solid tumor (diagnosis <8 weeks)	2 mandatory consults + educational sessions + exercise (aerobic + strength)	Education = 20′ + questions/debate and exercise = 60‘, 1×/week, 12 weeks	Usual care	QoL	EISPC	Education Exercise

SMAs: shared medical appointments, ESAS: Edmonton symptom assessment scale, CBT: cognitive behavioral therapy, GET: graded exercise therapy, KPS: Karnovsky performance scale, R: resistance, ECOG: ECOG performance status scale, EMG: electromyogram.

**Table 4 cancers-16-03371-t004:** Summary of findings.

PT versus Usual Care in Oncology Patients with Palliative Needs						
Population: Oncology Patients with Palliative Needs Intervention: PT Control: Usual Care						
Outcomes	Absolute Effects * (95% CI)		Relative Effect (95% CI)	Number of Participants (studies)	Certainty of the Evidence (GRADE)	Comments
	Risk with Usual Care	Risk with PT				
**Fatigue** Questionnaires FU: range 4 weeks to 13 months	-	-	Improved **	281 (4 RCTs)	⨁◯◯◯ Very low ^a,b,c^	A: ESAS, MFI-20, CSI-fatigue, EORTC-QLQ-C30 (v3.0) and SS PI: MBE, NMES, CBT, GET, nutritional counseling with a physical intervention S: Look et al. (2021) [26], Maddocks et al. (2013) [22], Poort et al. (2020) [25], and Uster et al. (2018) [23]
**Quality of Life** Questionnaires FU: range 1 week to 13 months	-	-	Improved **	1231 (6 RCTs)	⨁◯◯◯ Very low ^a,b,d^	A: FACITPC, McGill QoL Questionnaire, EORTC QLQ-30, LC-13, SIP8, and functional scale PI: educational and problem-solving sessions, massage, NMES, educational sessions combined with PT sessions, CBT, GET, nutritional counseling with a physical intervention S: Bakitas et al. (2009) [21], Kutner et al. (2008) [20], Maddocks et al. (2013) [22], Nottelmann et al. (2021) [27], Poort et al. (2020) [25], and Uster et al. (2018) [23]
**Nutrition (N1)** Questionnaires FU: range 0 days to 3 months	-	-	Improved **	98 (2 RCTs)	⨁◯◯◯ Very low ^b,c,e^	A: ESAS and three-day food diary PI: MBE and nutritional counseling with a physical intervention S: Look et al. (2021) [26] and Uster et al. (2018) [23]
**Nutrition (N2)** Objective measurements FU: 3 months	-	-	Contradiction between measurements	58 (1 RCT)	⨁◯◯◯ Very low ^c,e^	A: bioelectrical impedance analysis and weight scale PI: nutritional counseling with a physical intervention S: Uster et al. (2018) [23]
**Physical functioning (PHF1)** Questionnaires FU: range 4 weeks to 3 months	-	-	Improved **	192 (2 RCTs)	⨁◯◯◯ Very low ^a,b,c^	A: EORTC-QLQ-C30 and functional scale PI: CBT, GET, and nutritional counseling with a physical intervention S: Poort et al. (2020) [25] and Uster et al. (2018) [23]
**Physical functioning (PHF2)** Objective measurements FU: range 0 days to 3 months	-	-	Contradiction between studies	107 (2 RCTs)	⨁◯◯◯ Very low ^b,c,e^	A: manual muscle tester dynamometer, handgrip strength, 6-min walk test, and timed sit-to-stand test PI: NMES and nutritional counseling with a physical intervention S: Maddocks et al. (2013) [22] and Uster et al. (2018) [23]
**Psychosocial functioning** Questionnaires FU: range 2 weeks to 13 months	-	-	Improved ***	934 (5 RCTs)	⨁◯◯◯ Very low ^a,b,d^	A: CES-D, MPAC mood scale, ESAS, EORTC-QLQ-C30 and functional scale PI: educational and problem-solving sessions, massage, MBE, CBT, GET, and nutritional counseling with a physical intervention S: Bakitas et al. (2009) [21], Kutner et al. (2008) [20], Look et al. (2021) [26], Poort et al. (2020) [25], and Uster et al. (2018) [23]
**Pain** Questionnaires FU: range 7 days to 13 months	-	-	Improved ****	595 (5 RCTs)	⨁◯◯◯ Very low ^a,b,d^	A: NRS-11, pain intensity scale of MPAC, BPI, ESAS, BPI-T, SS and EORTC-QLQ-C30 v3.0 PI: scrambler therapy, massage, MBE, electromyography biofeedback assisted relaxation, and nutritional counseling with a physical intervention S: Kashyap et al. (2020) [24], Kutner et al. (2008) [20], Look et al. (2021) [26], Tsai et al. (2007) [21], and Uster et al. (2018) [23]

* The risk in the intervention group (and its 95% confidence interval) is based on the assumed risk in the comparison group and the relative effect of the intervention (and its 95% CI). CI: confidence interval; FU: follow up; A: assessment; PI: physiotherapy interventions; S: studies; MBE: mindful breathing exercise; NMES: neuromuscular electrical stimulation; CBT: cognitive behavioral therapy; GET: graded exercise therapy; ESAS: Edmonton symptom assessment scale; MFI-20: multidimensional fatigue inventory; EORTC-QLQ-C30: European Organisation for the Research and Treatment of Cancer Quality of Life Questionnaire Core 30; CSI-Fatigue: checklist individual strength, subscale fatigue severity; SS: symptom scale; FACITPC: functional assessment of chronic illness therapy for palliative care; QoL: quality of life; LC-13: lung cancer; SIP8: sickness impact profile; CES-D: Centre for Epidemiologic Studies depression; MPAC: memorial pain assessment card; BPI(-T): brief pain inventory (Taiwanese version). ** Half of the studies reported statistical significance (*p* < 0.05). *** Three out of five studies reported statistical significance (*p* < 0.05). **** One out of five studies reported statistical significance *p* < 0.05). GRADE Working Group grades of evidence. High certainty ◯◯◯⨁: we are very confident that the true effect lies close to that of the estimate of the effect. Moderate certainty ◯◯⨁◯: we are moderately confident in the effect estimate: the true effect is likely to be close to the estimate of the effect, but there is a possibility that it is substantially different. Low certainty ◯⨁◯◯: our confidence in the effect estimate is limited: the true effect may be substantially different from the estimate of the effect. Very low certainty ⨁◯◯◯: we have very little confidence in the effect estimate: the true effect is likely to be substantially different from the estimate of effect. Explanations (a–e). a. Lack of blinding was present and use of patient-reported outcomes, which are considered to be unvalidated outcome measures. b. The interventions took place in different countries, which may impact the way that these were carried out. This may impact the applicability of the interventions. c. The sample size is smaller than 400 participants. Besides, the confidence interval is not defined in the studies for this outcome. d. The confidence interval is not defined in more than half of the studies for this outcome. e. Lack of blinding was present.

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
