# Peer review of "Recommended Physiotherapy Modalities for Oncology Patients with Palliative Needs and Its Influence on Patient-Reported Outcome Measures: A Systematic Review"

_cancers, 2024, doi:10.3390/cancers16193371_

Round 1

Reviewer 1 Report

Comments and Suggestions for Authors

This manuscript presents a systematic review of physiotherapy modalities for oncology patients with palliative care needs, focusing on their impact on patient-reported outcome measures and provides a valuable contribution to the field, but could benefit from refinement in several sections to improve clarity, balance, and the emphasis on the quality of evidence:

Introduction: This section could benefit from more context on the current state of physiotherapy in palliative care oncology

MM: The authors indicate the end of the search period, but what was the initial period? Were all articles in the database included regardless of the date or was a specific period established? On the other hand , more information on how discrepancies between assessors were resolved could improve the clarity of this section

Results: This section is comprehensive but dense, a clearer narrative summarizing key findings alongside tables would improve readability. Table 4 presents a lot of information that may be overwhelming to the reader without a clear explanation in the text. A brief description highlighting key patterns in the data (e.g. which interventions consistently show positive effects, which outcomes had the strongest evidence) would make the table easier for the reader to understand.

Discussion: The limitations section could be expanded to address more potential biases (bias in the selected and/or included studies, attrition bias; some of the included studies have small sample sizes, which reduces the statistical power of their results; heterogeneity of interventions: Physiotherapy encompasses several techniques (e.g., cognitive behavioral therapy, graded exercise therapy, massage) that may make it difficult to generalize the findings. Expanding on this limitation would clarify that pooling the results of such diverse interventions weakens the ability to draw firm conclusions about their overall effectiveness which may be slightly exaggerated given the low quality of the evidence. The authors should put more emphasis on the need for higher quality research before making strong recommendations for clinical practice.

Author Response

Comment 1: Introduction: This section could benefit from more context on the current state of physiotherapy in palliative care oncology

Answer 1: Thank you for your valuable suggestion. Additional context on the current state of physiotherapy in palliative care oncology has been incorporated into the introduction, specifically on numbers 66 to 75. We believe this enhancement will provide readers with a clearer understanding of the topic.

Comment 2: MM: The authors indicate the end of the search period, but what was the initial period? Were all articles in the database included regardless of the date or was a specific period established? On the other hand , more information on how discrepancies be-tween assessors were resolved could improve the clarity of this section

Answer 2: Thank you for your insightful comments. The initial search period has been clarified in the methods section, specifying that all articles were included regardless of their publication date. This information can be found on numbers 107 to 110. Additionally, the resolution of discrepancies between assessors has been addressed, noting that a third independent reviewer was brought in to facilitate consensus.

Comment 3: Results: This section is comprehensive but dense, a clearer narrative summarizing key findings alongside tables would improve readability. Table 4 presents a lot of information that may be overwhelming to the reader without a clear explanation in the text. A brief description highlighting key patterns in the data (e.g. which interventions consistently show positive effects, which outcomes had the strongest evidence) would make the table easier for the reader to understand.

Answer 3: In response to the feedback regarding the density of the Results section, a clearer narrative summarizing key findings has been added on numbers 237-251, where a comprehensive summary is provided. This addition aims to enhance the readability of the data presented in Table 4 by highlighting key patterns and improving the overall understanding of the findings.

Comment 4: 

Discussion: The limitations section could be expanded to address more potential biases (bias in the selected and/or included studies, attrition bias; some of the included studies have small sample sizes, which reduces the statistical power of their results; heterogeneity of interventions: Physiotherapy encompasses several techniques (e.g., cognitive behavioral therapy, graded exercise therapy, massage) that may make it difficult to generalize the findings. Expanding on this limitation would clarify that pooling the results of such diverse interventions weakens the ability to draw firm conclusions about their overall effectiveness which may be slightly exaggerated given the low quality of the evidence. The authors should put more emphasis on the need for higher quality research before making strong recommendations for clinical practice.

Answer 4: In response to the feedback regarding the limitations section, we have indeed expanded our discussion to address additional potential biases, including those related to the selected studies, attrition bias, and the challenges posed by small sample sizes and heterogeneity of interventions. These enhancements can be found on numbers 307-327. This expansion clarifies how pooling results from diverse physiotherapy techniques can complicate generalizations and emphasizes the necessity for higher-quality research before making strong clinical recommendations.

Reviewer 2 Report

Comments and Suggestions for Authors

The introduction has established the background, relevance, and need for the review, clearly articulate its goals, and explain its significance in advancing the field. The authors provide a brief overview of cancer and its impact on patients health and discuss the role and potential benefits of physiotherapy for cancer patients. Regarding mentioning the situation of specific country, it is not necessary to include. Please removed Belgium references 4-5 and 8-9. the focus should remain on the broader context of the topic and the review's objectives.

The authors do not clarify the criteria used to order the articles included in the systematic review in Tables 2-3. We suggest ordering them by year of publication, from the most recent to the oldest.

When introducing acronyms, it's helpful to spell out the full name first, followed by the acronym in parentheses. Please explain the acronyms KNGF and EISPC.

Could the authors please provide the sources of funding for the studies included in the review?

In the QoL paragraph, the only RCT article with a low risk of bias is included. I recommend that the authors emphasize this point, as well as its findings.

Additionally, the authors should consider the risk of bias in individual studies when interpreting and discussing the results of the review. Moreover, the authors could explain and discuss whether any heterogeneity was observed in the results of the review, and if so, what type

Author Response

Comment 1: The introduction has established the background, relevance, and need for the review, clearly articulate its goals, and explain its significance in advancing the field. The authors provide a brief overview of cancer and its impact on patients health and discuss the role and potential benefits of physiotherapy for cancer patients. Regarding mentioning the situation of specific country, it is not necessary to include. Please removed Belgium references 4-5 and 8-9. the focus should remain on the broader context of the topic and the review's objectives.

Answer 1: We appreciate your feedback regarding the introduction. While we understand your suggestion to remove references specific to Belgium (references 4-5 and 8-9), we believe it is important to retain these citations as they highlight a clear issue: the integration of physiotherapy into palliative care. This aspect underscores the relevance of our review in addressing the challenges faced by cancer patients in various contexts. We aim to maintain a balance between the broader context of the topic and the specific challenges highlighted in our findings, all with due respect to your concerns.

Comment 2: The authors do not clarify the criteria used to order the articles included in the systematic review in Tables 2-3. We suggest or-dering them by year of publication, from the most recent to the oldest. 

Answer 2: Thank you for your suggestion regarding the organization of the articles in Tables 2 and 3. We appreciate your input and have indeed arranged the articles by year of publication, ordering them from the oldest to the most recent. This structure aims to provide a clearer perspective on the evolution of research in this area (144-150).

Comment 3: When introducing acronyms, it's helpful to spell out the full name first, followed by the acronym in parentheses. Please explain the acronyms KNGF and EISPC.

Answer 3: Thank you for your feedback regarding the introduction of acronyms. We have ensured that all acronyms, including KNGF (Koninklijk Nederlands Genootschap voor Fysiotherapie) and EISPC (Early Integrated Specialized Palliative Care), are spelled out in full followed by their respective abbreviations in parentheses. This should enhance clarity for all readers.

Comment 4: Could the authors please provide the sources of funding for the studies included in the review?

Answer 4: Thank you for your inquiry regarding the sources of funding for the studies included in our review. We would like to clarify that our review did not receive any funding. This is now also mentioned  in the manuscript (363).

Comment 5: In the QoL paragraph, the only RCT article with a low risk of bias is included. I recommend that the authors emphasize this point, as well as its findings.

Answer 5: We appreciate your feedback regarding the Quality of Life paragraph. We have taken your recommendation into account and emphasized that the only randomized controlled trial included with a low risk of bias has been highlighted, along with its significant findings (203-205).

Comment 6: Additionally, the authors should consider the risk of bias in individual studies when interpreting and discussing the results of the review. Moreover, the authors could explain and discuss whether any heterogeneity was observed in the results of the review, and if so, what type

Answer 6: Thank you for your valuable suggestion. We have addressed the risk of bias in individual studies while interpreting and discussing the results of the review. Additionally, we have provided an explanation regarding any observed heterogeneity in the results, including its nature, in the revised section from numbers 307 to 327.

Reviewer 3 Report

Comments and Suggestions for Authors

This paper states systematic review of recommended physiotherapy modalities for oncology patients with palliative needs and its influence on patient-reported outcome measures. It will help patients with palliative needs who like exercise and suffer from symptoms such as fatigue and pain. This is a suitable topic for Cancers readers interested in the effects of physiotherapy on cancer patients.

Major comments

1.  Tables 3 and 4 are very difficult to read. Would it be possible to add the results of each study to Table 3?

2.  The fact that the entire PRO is used as the outcome makes the table difficult to interpret. Couldn't you narrow down the topic a bit more? I understand the intention to know the overall effect of physiotherapy, though.

Minor comments

1.  It is better to write the full names of all the abbreviations in the table in the footnotes. Sometimes I don't understand what you mean when I read it, and I have to go back to the main text to understand what you mean.

Author Response

Comment 1: Tables 3 and 4 are very difficult to read. Would it be possible to add the results of each study to Table 3?

Answer 1: Thank you for your feedback regarding the readability of Tables 3 and 4. To enhance clarity and understanding, we have included a comprehensive summary of the results from each study in Table 3, spanning numbers 237 to 251. We hope this addition will make the information more accessible to readers.

Comment 2: The fact that the entire PRO is used as the outcome makes the table difficult to interpret. Couldn't you narrow down the topic a bit more? I understand the intention to know the overall effect of physiotherapy, though.

Answer 2: 

Thank you for your feedback regarding the use of the entire Patient-Reported Outcome (PRO) as the outcome measure in the table. It was indeed our intention to provide a comprehensive overview of the overall effects of physiotherapy rather than to narrow the focus. However, we acknowledge that this approach may lead to difficulties in interpretation.

In response to your concern, we have made necessary adjustments to enhance clarity and readability without losing the breadth of information. The aim was to balance the overall assessment of physiotherapy's impact while ensuring that the findings remain accessible and meaningful to the reader.

Round 2

Reviewer 1 Report

Comments and Suggestions for Authors

See L96, registration number is missing.

Reviewer 3 Report

Comments and Suggestions for Authors

Table 3 is now easier to read and understand as a whole. I also understood the author's intention of looking at the overall balance and seeing the benefits of physiotherapy. I think it has been sufficiently revised based on the comments.